# Improving Transfer Learning by means of Ensemble Learning and Swarm Intelligence-based Neuroevolution

**A. Gómez**[1,2]  **M. Abella**[1,2,3]  **M. Desco**[1,2,3,4]

[1]Bioengineering Department, Universidad Carlos III, Madrid, España
[2]Instituto de Investigación Sanitaria Gregorio Marañón, Madrid, España
[3]Centro Nacional Investigaciones Cardiovasculares Carlos III (CNIC), Madrid, España
[4]Centro de Investigación Biomédica en Red de Salud Mental (CIBERSAM), Madrid, España

**Abstract**  Neural Architecture Search (NAS) methods, when applied to very small but complex datasets, tend to overfit on the validation partitions and underperform compared to Transfer Learning models. In order to reduce the bias and variance of their predictions, Deep Ensemble Learning (DEL) can be used. The combination of NAS and DEL has only been employed on large datasets in the literature, but these scenarios do not present the overfitting in validation we typically experience, for instance, on medical imaging applications. In this work, we empirically assess the feasibility of NAS, DEL and the combination of the two on both small and large dataset scenarios. We find that the performance of the ensembles highly depend on the degree of overfitting of the standalone models.

## 1 Introduction

Most advances on Computer Vision are designed for and tested on large natural image datasets such as IMAGENET or CIFAR. Pretrained, highly performing models are readily available for the Deep Learning community, making Transfer Learning the standard approach for many research fields, such as medical image processing. However, despite their widespread popularity, the advantage of using these pretrained models on medical images has been questioned [Raghu et al. (2019), Neyshabur et al. (2020)]. The suitability of any given pretrained model has been shown to heavily rely on the inductive biases inherent to its architecture [Wang et al. (2019), Goyal and Bengio (2022)].

Selecting an adequate model for any specific application is not trivial, but several methods have been proposed to reduce bias and variance of sub-optimal models. One such method is Deep Ensemble Learning (DEL), which can achieve a high degree of complexity by combining the outputs of several relatively simple learners [Ganaie et al. (2022)]. Although powerful, DEL presents its own set of challenges. On the one hand, training several models requires considerably more time than fine-tuning a single architecture. On the other, it is difficult to obtain a high diversity on the features learnt by the individual learners if their architectures are identical and thus present the very same inductive biases.

Modifying the architecture of a pretrained Convolutional Neural Network is a complex, subjective and time-consuming endeavor because the optimal model hyperparameters are problem-dependent. Neural Architecture Search (NAS) techniques are used to explore the space of feasible models in an automatic way and have achieved state of the art performance on several computer vision benchmarks [White et al. (2023)]. However, balancing the exploration-exploitation trade-off is not trivial, especially when very deep networks are required: more layers imply more combinations between their hyperparameters, generating massive spaces to be explored. Moreover, considering a large number of architectures comes at the risk of overfitting on the validation partition of the datasets, which are used to select or discard the models. This problem is particularly relevant for

complex datasets where few samples are available as it may be very difficult to construct training and validation partitions that are representative of the whole task.

The problems faced by NAS can be mitigated with the use of DEL and vice versa. Ensemble methods could allow to limit the search to shallower models, considerably reducing the required computational expenses while maintaining the complexity of the prediction. Moreover, the feature sharing and weighting enhances the generalization capabilities of the models, overcoming the instances of overfitting on the validation partitions. Conversely, NAS methods already work with populations of solutions with distinct hyperparameters, which inherently present the high diversity that traditional DEL lacks. Combining the resulting models instead of selecting the best performing one might yield considerable benefits with negligible extra computational cost.

In this work, we have assessed the feasibility of DEL, NAS and a combination of the two to deal with the challenges of Transfer Learning in two completely different scenarios. The main goal is to assess the robustness of the methods against different degrees of overfitting. First, the different approaches were validated on the CIFAR-10 classification task. Several data and model weighting strategies were compared for the application of DEL. The NAS method employed was an in-house optimizer called the Chimera Algorithm. The combination of the Chimera Algorithm with DEL methods gave rise to a new, enhanced optimizer, referred to as the Gaggle Algorithm. All these methods were then employed to tackle a problem of biomedical interest to test their performance on a task where Transfer Learning would be the standard approach.

## 2 Related works

### 2.1 Deep Ensemble Learning

Ensemble Learning consists on training several individual learners on subsets of the original dataset and combine their predictions to leverage the influence of the most useful features learnt by each one of them [Breiman (2000)]. Deep Ensemble Learning (DEL) uses this strategy to enhance the predictive capabilities of Deep Learning models [Ganaie et al. (2022)]. There are many strategies to train Deep Learning models and combine them into ensembles, such as Bagging, Boosting, Stacking or Negative correlation Ensemble Learning. Particularly, Boosting Ensemble Learning methods train the different learners sequentially, based on the performance of the growing population. For each new learner, the dataset is either weighted or resampled in order to pay more attention to misclassified data. Each learner is also weighted based on its standalone performance, so that the final prediction is mainly driven by the most accurate ones. This approach allows for the use of shallower models that focus on specific parts of the task at hand rather than a single deep model that is more powerful on its own, but more prone to overfitting. However, some of the models present in the ensemble might not provide any valuable contribution, and even worsen the predictions. In order to limit this effect, ensemble pruning —also known as selective ensemble or ensemble selection— methods based on the validation error, kappa measure, complementary measure, margin, and diversity are commonly used [Margineantu and Dietterich (1997), Martínez-Muñoz and Suárez (2004), Guo et al. (2018)].

### 2.2 Neural Architecture Search

Neural Architecture Search (NAS) is a sub-field of metaheuristic optimization that strives to automatize the creation and optimization of Artificial Neural Networks. It spans a wide range of optimization algorithms, from the earliest works in the field employing Genetic Algorithms [Miller et al. (1989)] to modern approaches such as supernet optimization [Cha et al. (2023)], Multi-Fidelity MetaLearning [Zimmer et al. (2021)] or training Controller Recurrent Neural Networks using Reinforcement Learning to generate the architectures [Zoph and Le (2017)]. Each method presents their own strengths and drawbacks, but most of them are prone to overfit on the dataset partition used to select or discard the models, if it is not representative enough of the whole task. This

Table 1: Ensemble sample and model weighting strategies considered

| Sample weighting | Not weighting the data |
| | Resampling the dataset to only contain samples missclasified by the ensemble |
| | Resample the data randomly based on sample weight |
| | Modifying the loss function to take sample weight into cosideration |
| Model weighting | Averaging all the models' outputs |
| | Weighting each model's contribution on its individual validation accuracy or average loss |
| | Weighting each model's contribution for each output based on its validation confusion matrix or average loss for each one |

overfit is tangential to most methods, as it mainly depends on the distribution of our data and the model selection criterion employed [Cawley and Talbot (2010)]. Unfortunately, using rigorous model selection criteria within a NAS pipeline is oftentimes unfeasible due to computational costs. Combining the models generated through NAS into an ensemble has been shown to surpass the performance of the individual learners and is much less computationally expensive. However, the very few works that assess this approach in the literature [e.g. Herron et al. (2020), Chen et al. (2021)] only do so on large, widely studied datasets such as CIFAR-10, MNIST, IMAGENET or COCO. The behavior of ensembles attained through NAS in small and complex datasets, for which overfitting is a prevalent issue, is yet to be analyzed.

## 3  Materials and Methods

### 3.1  Algorithms

**3.1.1  Deep Ensemble Learning**. As previously stated, Boosting Ensemble methods allow for the use of shallower models. This is especially relevant in the context of NAS, as deeper models are much more computationally expensive to find —not just because of the higher training times, but because the possible hyperparameter combinations grow exponentially with the number of layers, generating a wider search space—. A modular Boosting Ensemble creator was developed, which sequentially trains copies of a given base model, weights their contributions to the growing ensemble, and modifies the training dataset according to the strategies specified by the user. In this work, we explored four strategies to modify the training dataset to assign a heavier weight to the samples that the ensemble is unable to properly characterize, as well as three strategies to calculate the contribution of each individual model to the complete ensemble. These are shown in Table 1. The sample weights were the ones employed by the AdaBoost classifier [Breiman (2000)]. The algorithm's pseudocode, default hyperparameters and explanation of the weights employed are depicted in Algorithm 1 in Appendix A.

**3.1.2  Neural Architecture Search**. The NAS method employed in this work is the Chimera Algorithm, a metaheuristic based on the Artificial Bee Colony Algorithm [Karaboga and Basturk (2007)] that was developed in-house. This algorithm deploys two kinds of optimizer agents —referred to as Employed and Onlooker Bees— on a population of candidate solutions. These agents select models from said population based on their individual performances, and explore similar architectures by using the flexible mutation operators employed in traditional Genetic and Evolutionary NAS methods. Unlike Genetic and Evolutionary approaches, though, the Chimera Algorithm is able to maintain a much higher diversity within the population —which is useful to obtain good performing ensembles— as the worst models are not discarded on each iteration, but rather simply given less

attention. Moreover, its local nature facilitates the exploration of similar architectures to that of any pretrained model provided by the user, exploring outwards from already promising positions within the search space. Its pseudocode is depicted in Algorithm 2 in Appendix A, along with a detailed algorithm description and a definition of its default hyperparameters.

### 3.1.3 Combination of DEL and NAS.
Both DEL and NAS approaches have been combined into the Gaggle Algorithm, which operates similarly to the Boosting Ensemble creator but calls the Chimera Algorithm to generate batches of optimized models instead of simply training a base architecture. Its pseudocode is depicted in Algorithm 3 in Appendix A.

## 3.2 Experimental Setup

The performance of DEL, NAS, and their combination were compared with that of predesigned architectures in two scenarios fully described in Sections 3.2.1 and 3.2.2. In the first scenario, these methods were evaluated on the classification problem of the natural images in the CIFAR-10 [Krizhevsky and Hinton (2009)] dataset. In the second scenario, they were tested on a problem of biomedical interest: the estimation of the horizontal misalignment of the detector in a Computed Tomography system by analyzing the artifacts present in the volumes reconstructed from a set of projections spanning an angle of 180º. In both scenarios, the Leslie Smith's learning rate test [Smith (2018)] was used to approximate the optimal learning rate for each model, as it proved itself much faster and reliable than adding an extra dimension to the hyperparameter space. Cross-validation has not been employed to compare individual models but, rather, to assess the validity of each method as a whole, generating a different split for each run to avoid biases due to the data partition used. The code developed can be accessed through this link: https://github.com/HGGM-LIM/Chimera. The seeds employed for randomization are specified in the code itself. An Intel® Core™ i7-7700 CPU and a NVIDIA® GeForce® RTX 2060 Super™ GPU were used for all tests performed.

### 3.2.1 Scenario I: natural image classification on CIFAR-10.
The following tests have been performed to evaluate the different approaches on the CIFAR-10 dataset, which consists on 60000 32x32 colour images divided into 10 diferent classes. 10000 images are reserved for testing and, in our setup, we used 90% of the remaining images for training, using the leftover ones for validation. No data augmentation was employed. All models were trained until the validation loss converged, with a patience of 8 epochs.

- One shallow and one deep feed-forward architectures were trained on the dataset to check the performance of standalone models. The shallow architecture selected was LeNet-5 [Lecun et al. (1998)] —which has around 60 thousand weights to optimize— and the deep architecture selected was VGG11 —over 133 million parameters, as reported in [Simonyan and Zisserman (2014)]—. In the case of VGG11, the last fully connected layer was modified to reduce the number of output classes from 1000 to 10. The weights of LeNet-5 were initialized at random, while VGG11 was tested both with random initialization and inheriting the weights from pretraining in ImageNet. The three cases were tested 32 times, generating a different training and validation partition of the dataset in each iteration. The AdamW optimizer was used. The learning rates employed were $10^{-3}$ and $10^{-4}$ for the shallow and deep models respectively, as suggested by the Leslie Smith test.

- LeNet-5 was used as a template architecture to create ensembles with. For this test, we employed all the data and model weighting strategies depicted in Table 1. Each new model was randomly initialized before training on the auxiliary dataset, which was obtained according to the data weighting scheme and the ensemble accuracies on the complete, unweighted dataset. Population pruning was performed after the whole ensembles were created. For simplicity, the models were selected simply based on their standalone validation error. Each data and model weighting combination was tested 10 times, reshuffling the training and validation partitions.

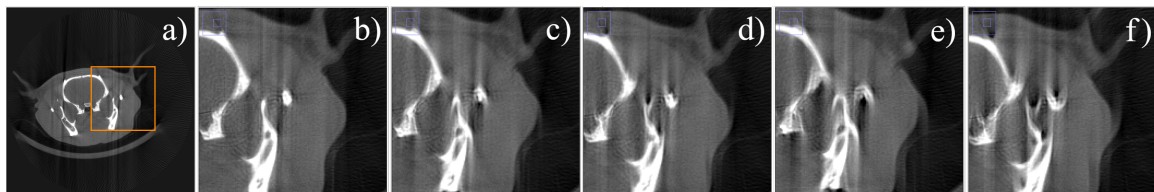

Figure 1: Axial view of a rodent head CT study reconstructed with an angular span of 180º and with no geometric misalignments (a). Detail of the region indicated in orange reconstructed with a simulated horizontal detector misalignment of 0mm (b), +0.5mm (c), -0.5mm (d), +1mm (e) and -1mm (f).

- The performance obtained with a simple NAS method was evaluated, using the Chimera Algorithm to create models from scratch. The models generated consisted of a feature extractor with convolutional and pooling layers and a single fully connected layer with 10 output classes. Two tests were performed. Firstly, it was run with a population size of 4 models and a search length of 16 iterations to check the capabilities of the algorithm to find good performing models in relatively shallow searches. Then, it was run with a population size of 8 models, a search length of 32 iterations and an exhaustion limit of 16 iterations to measure the possible improvement that a deeper search would yield. Each test was performed 4 times, reshuffling the training and validation partitions.

- The Gaggle Algorithm was employed to create ensembles with models generated from scratch. Once again, two distinct tests were performed. Firstly, the Gaggle Algorithm was run with 4 batches, a population size of 4 models, a search length of 16 iterations, accuracy-based model weighting and failed prediction data resampling in between batches. Then, the Gaggle Algorithm was run with a single batch, a population size of 8 models, a search length of 32 iterations and accuracy-based model weighting. Each test was performed 4 times, reshuffling the training and validation partitions.

### 3.2.2 Scenario II: misalignment regression on CT slices with artifacts.

Horizontal misalignment of the detector in a Computed Tomography system leads to artifacts in the reconstructed volumes, which can render the studies unusable for proper diagnosis. When the volumes are reconstructed with an angular span of 180º, these artifacts appear in the axial slices as arcs, as can be seen in Fig. 1. Their thickness depends on the misalignment value, and their orientation is given by the misalignment direction. In order to correct for these artifacts, a calibration file needs to be acquired manually by a technician. Theoretically, the geometrical misalignments of any acquisition could be automatically characterized based on the artifacts that appear on the pre-reconstructed volumes. Doing so would eliminate the need for manual calibration, and allow for the reconstruction of any study for which the callibration file is not available.

We generated a small dataset to train models able to predict the misalignment corresponding to any given individual slice. We first acquired six properly calibrated rodent cranial studies using a SEDECAL micro-CT system. Then, 25 sets of miscalibrated projections were simulated for each volume. Simulated horizontal misalignments values were taken from a random uniform distribution with a range of ±1mm. An FDK-based algorithm was used for reconstruction, generating volumes of 256×256×200 pixels, which were then normalized to ImageNet's mean and standard deviation —0.485 and 0.225, respectively— and separated into 2D axial slices. The tolerance, defined as the smallest misalignment that produces artifacts noticeable to the naked eye, was set to ±0.1 millimeters [Abella et al. (2021)]. The database was split into a training partition with five rodents, a validation partition with one rodent, and a test partition with one rodent as well —25000, 5000 and 5000 images, respectively—.

All models tested in this scenario were designed to work on individual slices, but the misalignment prediction for each volume was defined as the median of the predicted misalignments for all its slices. The use of the median reduced the influence of outliers coming from slices with less information. The Huber Loss [Huber (1964)] was used as the loss function for the prediction of the detector horizontal misalignment for every slice during training. The $\delta$ value in the Huber Loss, which controls the slackness for outliers, was selected via grid-search within a range [0, 0.5] in 0.1 intervals. The validation performance of VGG11 yielded 0.1 as the best $\delta$ value, which was consistent with the tolerance value defined and was thus employed for training all models. The following tests have been performed:

- Firstly, the shallowest and deepest pretrained models from the VGG Transfer Learning family proposed in [Simonyan and Zisserman (2014)], VGG11 and VGG19, were trained on the dataset defined. These were selected as they are some of the highest performing and most widely studied feedforward neural networks in image processing. The last fully connected layer of each classifier was modified to obtain a single regression value instead of the original 1000 class probabilities. The grayscale images were transformed into 3-channeled ones in order to match the expected input size. The AdamW optimizer was used with a cyclical learning rate whose bounds were selected by means of the Leslie Smith test with a cycle length of 4 epochs.

- Then, models were generated with the Chimera Algorithm in three different searches: (1) starting from scratch with a population size of 6 models and a search length of 16 iterations, (2) starting from 4 mutated copies of the VGG11 model and a search length of 12 iterations, and (3) starting from 4 mutated copies of the VGG19 model and a search length of 8 iterations. The population sizes and search lengths were selected such that all runs would take around 48 hours to complete on our setup. No exhaustion limit was specified in any of the three cases. In all cases, only the feature extractors were mutated, keeping the classifiers constant as three fully connected layers of 4096, 4096, and 1 output features respectively with ReLU and dropout layers with a probability of 0.5 in between. The AdamW optimizer was used with a cyclical learning rate with a cycle length of 4 epochs as well. Suitable bounds for the learning rates had to be calculated before training every model using the Leslie Smith test, as in this case it did not produce stable learning rates for differently sized randomly generated models. To reduce the time required to obtain such learning rates, the search was restricted to an order of magnitude above and below those of the parent model to the one to be trained, relying on the assumption that small changes in the architecture would not yield great fluctuations in the optimum learning rate bounds. The Chimera Algorithm's output models are referred to as $Scratch_{NAS}$, $VGG11_{NAS}$ and $VGG19_{NAS}$.

- Each of the three populations generated with the Chimera Algorithm were combined into ensembles by means of the Gaggle Algorithm. These were pruned based on the models' Mean Absolute Error (MAE) in the median prediction of the validation volumes, generating an ensemble out of each population. These ensembles are referred to as $Scratch_{DEL}$, $VGG11_{DEL}$ and $VGG19_{DEL}$. Moreover, in order to palliate the effect of the simple ensemble pruning strategy employed, all model combinations within each population have been tested to check the theoretical performance of an optimal ensemble generated from each population. The models were weighted based on their validation MAE, but the best ensembles were selected based on their test MAE. These optimum ensembles are referred to as $Scratch_{Best}$, $VGG11_{Best}$ and $VGG19_{Best}$.

All methods —Transfer Learning, the Chimera and the Gaggle Algorithms— have been tested twice, selecting a different validation rodent for each of the two cross-validation partitions.

Table 2: Results of the methods proposed on the CIFAR-10 dataset. The shallow searches were carried out with a population size of 4 models and a search length of 16 iterations, while the deep ones were carried out with a population size of 8 models and a search length of 32 iterations. All tests performed, including all ensemble approaches, required a total of 1077.86 GPU hours

| Approach | | Training % accuracy | Validation % accuracy | Test % accuracy | Computing time (h) | Number of models |
|---|---|---|---|---|---|---|
| Base models | LeNet-5 | 68.96 ± 1.73 | 60.52 ± 0.76 | 60.52 ± 0.83 | 0.17 ± 0.04 | 1 |
| | VGG11 (random init.) | 85.43 ± 3.43 | 70.76 ± 0.98 | 70.56 ± 0.85 | 0.24 ± 0.03 | 1 |
| | VGG11 (pretrained) | 93.87 ± 1.61 | 84.39 ± 0.87 | 84.01 ± 0.61 | 0.23 ± 0.01 | 1 |
| DEL | Acc-based weightings | 80.83 ± 0.72 | 71.01 ± 0.57 | 70.83 ± 0.30 | 5.43 ± 0.15 | 25.00 ± 1.26 |
| NAS | Shallow searches | 85.67 ± 2.84 | 74.52 ± 0.91 | 74.43 ± 0.97) | 8.04 ± 0.23 | 1 (5.2 ± 1.3 generated) |
| | Deep search | 85.07 ± 4.73 | 74.08 ± 4.02 | 73.74 ± 4.00 | 36.23 ± 6.06 | 1 (23.0 ± 4.3 generated) |
| DEL + NAS | Shallow searches | 94.87 ± 0.45 | 81.76 ± 0.73 | 81.11 ± 0.25 | 36.05 ± 1.78 | 14.75 ± 2.86 |
| | Single deep search | 94.35 ± 1.08 | 81.61 ± 0.84 | 81.05 ± 0.84 | 36.62 ± 5.79 | 19.00 ± 6.68 |

## 4 Results

The main results obtained by the previously discussed approaches are presented for the CIFAR-10 image classification task and the CT geometrical misalignment characterization task in Sections 4.1 and 4.2, respectively. The complete behavior of all approaches is laid out in Appendix B.

### 4.1 Results for the Scenario I

The accuracies attained in the CIFAR-10 dataset with all methods proposed are shown in Table 2. Regarding the standalone predesigned architectures, VGG11 attained better accuracies in training, validation, and testing throughout cross-validation.

The behavior of an example ensemble employing accuracy-based data resampling and model weighting is shown on Fig. 2. The ensemble accuracy increased as more models were added, although the rate of improvement decayed rapidly. The final ensemble was pruned, further improving the ensemble performance up until fundamental models started to get removed and the accuracy lowered back to that of the best standalone one. In all cases, the pruned populations produced better results than the original ones, using less models. The maximum training, validation and test accuracies were obtained when using accuracy-based data resampling and accuracy-based model weighting, which are shown in Table 2. For the complete results of all data and model weighting strategies, refer to Table 4 in Appendix B.

The best models in the NAS setting were always found in the deep searches. However, some low performing models also appeared in deep searches due to the late exhaustion and reinitialization of some solutions, which lowered their mean population accuracy. The time required and the number

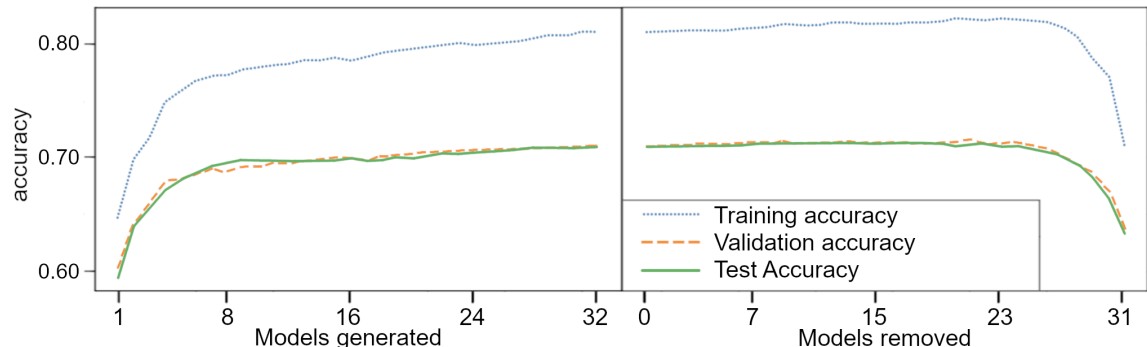

Figure 2: Performance of an example ensemble, adding the models by creation order (left) or removing them according to their standalone accuracies (right).

Table 3: Results of the methods proposed on the CT artifacts dataset. All tests performed required a total of 283.52 GPU hours. The best performing approaches according to the test Mean Absolute Error (MAE) are highlighted in bold.

| Model | Cross-validation I | | Cross-validation II | |
| --- | --- | --- | --- | --- |
| | Validation MAE (mm) | Test MAE (mm) | Validation MAE (mm) | Test MAE (mm) |
| VGG11 | 0.0603 | 0.1890 | 0.1204 | 0.1108 |
| VGG19 | **0.0377** | **0.0672** | **0.0638** | **0.0796** |
| Worst Scratch$_{NAS}$ | 0.0134 | 0.3335 | 0.1070 | 0.2488 |
| Best Scratch$_{NAS}$ | **0.0121** | **0.2915** | **0.0520** | **0.1914** |
| Scratch$_{DEL}$ | 0.0131 | 0.3123 | 0.0466 | 0.1927 |
| Scratch$_{Best}$ | 0.0129 | 0.3007 | 0.0467 | 0.2005 |
| Worst VGG11$_{NAS}$ | 0.0854 | 0.1814 | 0.0141 | 0.0344 |
| Best VGG11$_{NAS}$ | **0.0399** | **0.1559** | 0.0158 | 0.0293 |
| VGG11$_{DEL}$ | 0.0436 | 0.1674 | 0.0126 | 0.0297 |
| VGG11$_{Best}$ | 0.0496 | 0.1603 | **0.0148** | **0.0283** |
| Worst VGG19$_{NAS}$ | 0.0342 | 0.2067 | 0.0676 | 0.1279 |
| Best VGG19$_{NAS}$ | **0.0244** | **0.1728** | 0.0719 | 0.0378 |
| VGG19$_{DEL}$ | 0.0176 | 0.1857 | 0.0670 | 0.0804 |
| VGG19$_{Best}$ | 0.0216 | 0.1791 | **0.0706** | **0.0281** |

of models generated in the deep searches were approximately 4-fold that of the shallow searches, which was consistent with the batch size and search length.

Combining NAS with DEL produced better results than any of the two methods on their own, but the ensembles were not able to surpass the accuracy of the pretrained VGG11. Using the Gaggle Algorithm with failed prediction data resampling in between several shallow Chimera searches produced slightly more consistent results than simply combining the results of a single long search.

## 4.2 Results for the Scenario II

The most representative results for each method in the CT artifacts scenario are shown in Table 3 for both cross-validation partitions. The performance of the models and ensembles was measured by the MAE of their volume-wise predictions. The best and worst generalizing models generated from each search are shown, as well as the ensembles —composed of at least two models— generated by pruning and the best possible model combination. The complete results are reported in Appendix

B. VGG19 performed better than VGG11, and starting the search from either of the two resulted in considerably better performance than starting from scratch. All the ensembles generated with the Gaggle Algorithm outperformed the worst model of their population in test, but none outperformed the best one.

## 5 Discussion and conclusions

Each method considered on the CIFAR-10 dataset attained a very similar accuracy in the validation and test partitions, as shown in Table 2. The validation partition was representative enough of the whole task, and thus the validation accuracy could be used both for early stopping and as a model selection criterion, with no signs of overfitting. The importance of pretraining the models is highlighted by the significant difference in the accuracies attained by VGG11 depending on the weight initialization strategy employed.

The accuracy of the LeNet-5 ensembles matched that of the randomly initialized VGG11, and the NAS searches commencing from scratch surpassed both of them. The combination of DEL and NAS performed consistently better than either standalone method, which is consistent with the literature [Herron et al. (2020), Chen et al. (2021)]. Combining the models required negligible computational requirements, the resampling of the training dataset made each epoch slightly shorter and, especially, allowed to limit the search to very shallow models which, when combined, almost rivaled the accuracy of the pretrained VGG11. This was no trivial achievement, considering that the models were created from scratch, with no data augmentation, learning rate scheduler, nor a priori guidance, using a simple NAS algorithm and in under two days of computing.

Pruning was always necessary to achieve the best performing ensembles, which makes sense particularly when the learners were created via NAS because they presented a much higher diversity —and performance— within the population. Thus, further research onto how the models are combined —to only extract valuable contributions from each model, for instance by employing a Mixture of Experts approach [Chen et al. (2022)]— and trained —to avoid the creation of uninformative models whatsoever— is crucial to further improve the performance of our systems while at the same time reducing the time and resources needed to create them.

On the other hand, the models trained on the CT misalignments dataset presented a much greater tendency to overfit on the validation partition, as evidenced by the order of magnitude difference between validation and test performances in most cases throughout Table 3. This occurred due to the relatively small dataset size compared to the problem complexity. The models trained on this dataset presented a great variance in their predictions because the validation partition failed to characterize the whole problem [Cawley and Talbot (2010)]. In most cases, all the models generated within a NAS run performed similarly, attaining much better validation than test MAE. In these cases, their combination into ensembles, even with the theoretical optimal pruning, attained similar or even worse results. In the second cross-validation partition, there was a significant difference between the overfitting presented by some of the VGG19$_{NAS}$. This led to the VGG19$_{DEL}$ generated to perform considerably worse than the best VGG19$_{NAS}$ on its own. In all cases, the models that overfit the most are introduced first and their contributions are given more weight due to their higher validation accuracy.

From all this, we derive that the combination of the models generated into ensembles based on their validation performance seems to be beneficial when there is a low risk of overfitting. If that is not the case, there are two possible outcomes. In the best case scenario, all models present a similar degree of overfitting, and their combination performs similarly. Otherwise, we might add models to the ensemble that smear the contribution of better generalizing ones, giving more weight to their contributions due to their higher performance in validation. One option to perform a well-founded ensemble pruning is to consider the uncertainty of the models predictions, as quantified through methods such as those explored in the literature [Abdar et al. (2021)]. Moreover, the ensembles could be pruned based on the performance of the models on meta-validation dataset partitions, given

that enough data is available. Despite not always outperforming the best individual learners, all ensembles generated generalize considerably better than the worst models on their own. Reducing the variance of the models via DEL might still prove itself a sensible approach for small dataset scenarios, considering that the test partition can be as biased as the validation one.

All of these observations are carried out in two very specific scenarios, though. A single NAS method is employed, modifying only the feature extractors of feedforward Convolutional Neural Networks. No data augmentation approaches are used, and the analysis is performed on just two datasets that, although representative of the kind of problems they present, are not sufficient to characterize the wide NAS use case. The interaction with the many techniques employed in the Deep Learning workflow might vary the behavior of the ensembles and the optimal way of combining and weighting the models. Moreover, there is very limited knowledge about how to select the best NAS, DEL and ensemble pruning method for any given problem, requiring significant expertise from the final user. We call for further research on use cases such as the one presented herein, for which the Computer Vision tools at our disposal are not yet applicable.

On the bright side, given that the data available is either representative of the whole task or abundant enough as to make independent model selection and ensemble pruning partitions, the combination of DEL and NAS appears to be a promising strategy. DEL can be integrated almost effortlessly to any population-based NAS framework, with negligible extra computation costs to the final user. Moreover, these methods require minimal human intervention, which makes them perfect for settings where Deep Learning experts are rarely available —like biomedical laboratories—. Coincidentally, the kind of data produced and processed in these settings tend to differ greatly from that of large natural images datasets for which powerful Transfer Learning models are readily available.

In conclusion, we found that the combination of populations generated through NAS methods into ensembles only generalize better than the standalone models provided that they all present a similar degree of overfitting. Such is not always the case for models generated via NAS using very small datasets, for which building representative validation and test partitions is not trivial. In these scenarios, DEL can mitigate the influence of the worst models generated, but it does not ensure the best generalization to the test datasets. Sensible ensemble pruning must be carried out to identify and discard the heavily overfitting models in order to achieve the most out of the generated populations.

## 6 Broader Impact Statement

We have shown that the combination of the resulting models from a Neural Architecture Search into Deep Ensembles can significantly increase their generalization capabilities, even in scenarios where the low amount of data yields an overfitting on the validation partitions. This allows the use of AutoML in scenarios where they were previously unfeasible. However, automation bias might lead non-experts in Deep Learning to fall for the luring generalization of the resulting models to the test datasets, i.e. trusting their predictions with no regards for their interpretability. Said interpretability, which is crucial in high-stakes scenarios such as medical imaging, is generally lost upon the combination of the individual models into ensembles [Kook et al. (2022)]. Advancements on the AutoML field must be taken with caution until we are able to reliably distinguish sensible architectures from fortuitous nonsense.

On the other hand, AutoML methods benefit greatly from higher computational resources, leading to a monopolization of AutoML by those with access to high computing power. Although this holds true for our Deep Ensembles scenario, the methods proposed herein allow to obtain feasible results while limiting the search to shallower models. This could reduce the computational resources required which, in turn, can both mitigate the environmental impact of AutoML and help democratize the application of Deep Learning in low-budget scenarios.

**Acknowledgements**. This work was supported by Ministerio de Ciencia e Innovación, Agencia Estatal de Investigación and co-funded by the European Regional Development Fund, 'A way of making Europe': PID2019-110369RB-I00/AEI/10.13039/501100011033 (RADHOR); PDC2021-121656-I00 (MULTIRAD), funded by MCIN/AEI/10.13039/501100011033 and by the European Union 'NextGenerationEU'/PRTR. Also funded by Instituto de Salud Carlos III through the projects PT20/00044, co-funded by the European Regional Development Fund "A way to make Europe" and PMPTA22/00121 and PMPTA22/00118, co-funded by the European Union 'NextGenerationEU'/PRTR. The CNIC is supported by Instituto de Salud Carlos III, Ministerio de Ciencia e Innovación, and the Pro CNIC Foundation.

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

# A Appendix A: Algorithms

All algorithms described in this section can be accessed through this link: https://github.com/HGGM-LIM/Chimera.

---

**Algorithm 1**: Ensemble Generator

| | | |
|---|---|---|
| **Input:** | | $t\_loader$, a data loader containing the training partition of the dataset |
| | | $v\_loader$, a data loader containing the validation partition of the dataset |
| | | $base\_model$, the base architecture to create the ensembles with |
| | | $n\_models$, the ensemble length |
| **Output:** | | $ensemble$, a list containing all trained models |
| | | $weights$, matrix with the weights of each learner for each target dimension |

1   $ensemble \leftarrow \emptyset$
2   $weights \leftarrow \emptyset$
3   $t\_loader_{aux} \leftarrow t\_loader$
4   **for** $i \leftarrow 1$ **to** $n\_models$ **do**
5      model $\leftarrow$ **train**($base\_model, t\_loader_{aux}$)
6      $ensemble \leftarrow ensemble \cup model$
7      $weights \leftarrow$ **CalculateWeights**($model, v\_loader$) $\cup model$
8      $t\_loader_{aux} \leftarrow$ **ModifyDataset**($t\_loader, ensemble, weights$)
9   $ensemble, weights \leftarrow$ **pruneEnsemble**($ensemble, weights, v\_loader$)

---

The ensemble generator depicted here is a modular Boosting Ensemble algorithm. The functions **CalculateWeights** and **ModifyDataset** follow any of the strategies depicted in Table 1. The **pruneEnsemble** function orders the models based on their validation accuracy and discards the worse ones until the ensemble performance starts decreasing.

The sample weights are based on the ones employed by the AdaBoost [Breiman (2000)] classifier. In classification problems, these weights are given by $\exp(-\alpha \cdot acc)$ if the datapoint has been properly classified by the ensemble, or $\exp(\alpha \cdot acc)$ if it has not, where $acc$ is the complete ensemble's training accuracy and $\alpha$ is a hyperparameter that controls how much misclassified samples are penalized; in regression ones, it will be given by $\exp(-\alpha \cdot loss_v)$, where $loss_v$ is the validation loss. By default, $\alpha$ is set to 1.

The Chimera Algorithm is an in-house Neural Architecture Search method that follows the same workflow as the Artificial Bee Colony Algorithm [Karaboga and Basturk (2007)]. The population of solutions can be initialized by copying a model provided or by generating architectures with a random —up to a maximum defined— amount of layers, depending on the $init\_args$ passed. Then, two types of optimizer agents, referred to as $Bees_{Employed}$ and $Bees_{Onlooker}$, will take turns to explore the hyperparameter space until $stop\_criterion$ is met. The $Bees_{Employed}$ bind to each model in a one-to-one assignment (line 8) while the $Bees_{Onlooker}$ randomly choose a model every iteration with a probability proportional to its validation performance (line 27). Some models might not be selected in this step, whereas others could be selected by several $Bees_{Onlooker}$ within a single iteration. This drives the exploration-exploitation tradeoff to lean towards the latter for the most promising regions in the hyperspace.

The two types of $Bee$ explore around their selected model by duplicating it and performing a series of mutation steps on the copy. We keep an $exhaustion$ counter for each solution that increases every time the mutated model is worse than the original one, and resets to zero when it is not. If a solution's exhaustion counter exceeds a certain threshold it is saved as a plausible global minimum and its Employed Bee reassigns itself to a newly initialized solution (lines 20-25).

---

**Algorithm 2:** Chimera Algorithm

---

**Input:**     $t\_loader$, a data loader containing the training partition of the dataset

                $v\_loader$, a data loader containing the validation partition of the dataset

                $Np$, the population size

                $stop\_criterion$, a condition to stop searching for models

                $ex_{max}$, the threshold for the exhaustion counter to consider the solution exhausted

                $init\_args$, arguments for the model initialization

                $mut\_args$, arguments for the model mutation

**Output:** $final\_models$, a list containing all trained models

1   Generate a $Bees_{Employed}$ population and a $Bees_{Onlooker}$ one of size $Np$ each

2   $final\_models \leftarrow \emptyset$

3   **for** $i \leftarrow 1$ **to** $Np$ **do**

4      $models_i \leftarrow \textbf{\textit{initialize}}(init\_args)$

5      $models_i \leftarrow \textbf{\textit{train}}(models_i, t\_loader)$

6      $score_i \leftarrow (models_i, v\_loader)$

7      $exhaustion_i \leftarrow 0$

8      $models_i \xleftarrow{\text{binds to}} Bees_{Employed,i}$

9   **while** $stop\_criterion$ is not met **do**

10      **for** $Bee_{Employed,i} \in Bees_{Employed}$ **do**

11         $new\_model \leftarrow \textbf{\textit{mutate}}(models_i, mut\_args)$

12         $new\_model \leftarrow \textbf{\textit{train}}(new\_model, t\_loader)$

13         $score_{new} \leftarrow \textbf{\textit{score}}(new\_model, v\_loader)$

14         **if** $score_{new} < score_i$ **then**

15            $models_i \leftarrow new\_model$

16            $score_i \leftarrow score_{new}$

17            $exhaustion_i \leftarrow 0$

18         **else**

19            $exhaustion_i + = 1$

20         **if** $exhaustion_i \geq ex_{max}$ **then**

21            $final\_models \leftarrow final\_models \cup model_i$

22            $models_i \leftarrow \textbf{\textit{initialize}}(init\_args)$

23            $models_i \leftarrow \textbf{\textit{train}}(models_i, t\_loader)$

24            $score_i \leftarrow \textbf{\textit{score}}(models_i, v\_loader)$

25            $exhaustion_i \leftarrow 0$

26      **for** $Bee_{Onlooker,j} \in Bees_{Onlooker}$ **do**

27         $model_j \xleftarrow{\text{is chosen from}} models$ with probability $\propto scores$

28         $new\_model \leftarrow \textbf{\textit{mutate}}(model_j, mut\_args)$

29         $new\_model \leftarrow \textbf{\textit{train}}(new\_model, t\_loader)$

30         $score_{new} \leftarrow \textbf{\textit{score}}(new\_model, v\_loader)$

31         **if** $score_{new} < score_j$ **then**

32            $model_j \leftarrow new\_model$

33            $score_j \leftarrow score_{new}$

34            $exhaustion_j \leftarrow 0$

35         **else**

36            $exhaustion_j + = 1$

37

---

The models are trained until convergence in order to properly compare their performance on the validation partition. Each model generated is then assigned a score based on the Artificial Bee Colony fitness value, given by $(1 + loss_{valdation})^{-1}$. This score is used to compare the original and mutated models and discard the worst performing one. The $Bees_{Onlooker}$ will also use this score to select models.

The hyperparameters of the Chimera Algorithm are: the condition to stop the search $stop\_criterion$ —either a maximum number of iterations or a threshold for the objective loss function—, the population size $Np$, the model exhaustion limit $ex_{max}$, the arguments for the **initialize** function $init\_args$, and the arguments for the **mutate** function $mut\_args$. The latter comprises the number of mutations per **mutate** call, the probability of performing each mutation type —that is, adding, subtracting or modifying each kind of layer—, and, optionally, the bounds to the model hyperparameters' space —the maximum number and types of layers, kernel sizes or strides, or the range of learning rates—.

The default hyperparameters are defined as follows. The number of mutations performed on each exploration attempt is defined as $|N(1, \sqrt[3]{1 + exhaustion_i})|$ rounded upwards, to ensure that most of the time we perform very few mutation steps. Bigger steps to overcome local minima are only allowed if the solution is close to exhaustion. In this way, we make sure that the close neighborhood of a given solution is properly exploited before trying to reach further away. The probabilities of adding, deleting, or mutating a layer on each exploration step are set to 30% each, while there is a 10% probability of simply resetting the weights of some layers while leaving the architecture intact. When adding or mutating a layer, its probability of being a convolutional layer was set as $5n_p/(5n_p + n_c)$, where $n_p$ is the number of pooling layers and $n_c$ that of convolutional ones in the model to mutate. Otherwise, the new layer is a pooling one, with equal probability of being either max or average pooling. This ensures that our models will tend to present 5 times as many convolutional layers as pooling ones. Convolutional and pooling kernel sizes are drawn from a uniform distribution from 1 to 7, and the model length is only limited by the available GPU memory. When adding a convolutional layer, the input and output channels are given by those of the surrounding layers. By default, when mutating a convolutional layer, the number of output channels —and thus input channels of the next convolutional layer— can vary from half to twice the original number of channels. Model exhaustion is deactivated by default, with $ex_{max}$ being infinity. If no base architecture is provided, the models will be initialized at random within the search bounds. There are no default $stop\_criterion$ as this is specific to every problem and available resources.

The stopping criteria and the search bounds are the hyperparameters that most significantly affect the quality of the models produced. The initialization arguments, probability of each mutation type, population size and model exhaustion limit simply define the starting point, preferred direction and average search speed throughout the hyperspace. For instance, 1) keeping a higher probability of removing layers rather than adding them yields a search focused on decreasing model complexity, or 2) using a small population size with a high exhaustion limit favors a thorough exploitation of a few regions rather than widespread exploration. The optimal hyperparameters will be problem-dependent, and could be either fixed or adaptive, leveraging a-priori knowledge provided with the quality of the models found throughout the search.

| **Algorithm 3:** Gaggle Algorithm |
|---|

| **Input:** | $t\_loader$, a data loader containing the training partition of the dataset |
|---|---|
| | $v\_loader$, a data loader containing the validation partition of the dataset |
| | $Np$, the population size |
| | $n\_batches$, the number of times the Chimera Algorithm is called |
| | $iter_{max}$, a maximum number of iterations to run the search for |
| | $loss_{min}$, a threshold for the loss value to stop the search when achieved |
| | $ex_{max}$, the threshold for the exhaustion counter to consider the solution exhausted |
| | $init\_args$, arguments for the model initialization |
| | $mut\_args$, arguments for the model mutation |
| **Output:** | $ensemble$, a list containing all trained models |
| | $weights$, matrix with the weights of each learner for each target dimension |

1   $ensemble \leftarrow \emptyset$
2   $weights \leftarrow \emptyset$
3   $t\_loader_{aux} \leftarrow t\_loader$
4   **for** $i \leftarrow 1$ **to** $n\_models$ **do**
5      $model\_batch \leftarrow$
      **ChimeraAlgorithm**$(t\_loader_{aux}, v\_loader, Np, iter_{max}, loss_{min}, ex_{max}, init\_args, mut\_args)$
6      **for** $model \in model\_batch$ **do**
7         $weights \leftarrow weights \cup$ **CalculateWeights**$(model, v\_loader)$
8      $ensemble \leftarrow ensemble \cup model\_batch$
9      $ensemble, weights \leftarrow$ **pruneEnsemble**$(ensemble, weights, v\_loader)$
10     $t\_loader_{aux} \leftarrow$ **ModifyDataset**$(t\_loader, ensemble, weights)$

The Gaggle Algorithm is an extension of the Chimera Algorithm that retrieves the populations of models generated and adds them to a growing ensemble. The ensemble is pruned with the addition of each new batch of models. The **CalculateWeights**, **ModifyDataset** and **pruneEnsemble** functions work in the same way as in Algorithm 1.

## B   Appendix B: Complementary results

The performance of the LeNet-5 ensembles generated using all the combinations of sample weighting —no data weighting, DN; failed prediction resampling, DF; accuracy-based resampling, DA; loss function modification, DL— and model weighting —mean prediction, MM; accuracy-based weighting, MA; confusion matrix-based weighting, MC— depicted in Table 1 are shown in Table 4. Both the complete and pruned populations —C and P, respectively— are shown. The pruned ensembles always attained better results than the original ones. The best results were obtained by a very slight margin with accuracy-based data resampling and accuracy-based model weighting, highlighted in bold. The only ensembles that perform noticeably worse in validation and test are the ones employing the loss function modification strategy, highlighted in grey.

The complete results on the CT artifacts scenario are shown in Tables 6 and 5. The models generated with the Chimera Algorithm are ordered and numbered based on their Mean Absolute Error on the validation dataset. The ensembles generated with the Gaggle Algorithm are ordered and numbered based on the number of models they contain. In the first cross-validation partition, the only model that did not overfit was VGG19, while neither Transfer Learning models nor most of their evolutions and ensembles presented overfit in the second cross-validation partition. The best Transfer Learning model is highlighted in blue. The best Chimera output model for each run based

Table 4: Results of the methods proposed on the CIFAR-10 dataset

| Model w. / Data w. | | MM Train acc. | MM Valid. acc. | MM Test acc. | MA Train acc. | MA Valid. acc. | MA Test acc. | MC Train acc. | MC Valid. acc. | MC Test acc. |
|---|---|---|---|---|---|---|---|---|---|---|
| DN | C | 79.73 ±0.49 | 70.46 ±0.48 | 70.30 ±0.28 | 80.11 ±0.73 | 70.74 ±0.64 | 70.42 ±0.49 | 79.44 ±0.87 | 69.88 ±0.54 | 69.87 ±0.43 |
| DN | P | 80.20 ±0.52 | 70.71 ±0.45 | 70.45 ±0.29 | 80.61 ±0.82 | 70.94 ±0.61 | 70.61 ±0.46 | 79.79 ±0.90 | 70.05 ±0.51 | 70.01 ±0.39 |
| DF | C | 80.23 ±0.78 | 70.75 ±0.33 | 70.54 ±0.40 | 80.46 ±0.66 | 70.84 ±0.61 | 70.68 ±0.28 | 79.33 ±0.63 | 69.90 ±0.41 | 69.91 ±0.43 |
| DF | P | 80.46 ±0.76 | 70.85 ±0.35 | 70.63 ±0.38 | **80.83 ±0.72** | **71.01 ±0.57** | **70.83 ±0.30** | 79.83 ±0.78 | 70.13 ±0.46 | 70.10 ±0.42 |
| DA | C | 79.81 ±0.55 | 70.40 ±0.49 | 70.41 ±0.37 | 80.04 ±0.55 | 70.90 ±0.51 | 70.48 ±0.30 | 79.11 ±0.64 | 70.17 ±0.47 | 70.06 ±0.35 |
| DA | P | 80.18 ±0.65 | 70.59 ±0.46 | 70.58 ±0.32 | 80.35 ±0.56 | 71.06 ±0.56 | 70.65 ±0.37 | 79.63 ±0.58 | 70.36 ±0.49 | 70.18 ±0.39 |
| DL | C | 79.57 ±1.13 | 66.82 ±0.53 | 66.58 ±0.43 | 79.93 ±0.86 | 66.89 ±0.506 | 66.70 ±0.332 | 77.97 ±1.38 | 66.96 ±1.24 | 66.64 ±1.60 |
| DL | P | 79.63 ±1.10 | 66.90 ±0.57 | 66.63 ±0.45 | 79.94 ±0.91 | 66.98 ±0.55 | 66.71 ±0.37 | 78.29 ±1.36 | 67.21 ±1.29 | 66.83 ±1.56 |

on their test Mean Absolute Error is highlighted in magenta, and the best Gaggle one is highlighted in orange. The best possible ensembles to be created with each population are highlighted in green.

A visual representation of the Transfer Learning models and the architectures generated in the first cross-validation partition are shown in Figures 3, 4 and 5.

Table 5: Results on the first cross-validation partition of the CT misalignments dataset

| Model | Training Absolute Error (mm) | Validation Absolute Error (mm) | Test Absolute Error (mm) |
|---|---|---|---|
| VGG11 | 0.0950±0.0557 | 0.0603±0.0518 | 0.1890±0.1412 |
| VGG19 | 0.0326±0.0241 | **0.0377±0.0323** | **0.0672±0.0748** |
| Scratch$_{NAS}$1 | 0.0267±0.0152 | **0.0121±0.0091** | **0.2915±0.0967** |
| Scratch$_{NAS}$2 | 0.0273±0.0206 | 0.0134±0.0088 | 0.3335±0.0920 |
| Scratch$_{NAS}$3 | 0.0177±0.0211 | 0.0151±0.0083 | 0.3104±0.0843 |
| Scratch$_{NAS}$4 | 0.0232±0.0180 | 0.0154±0.0108 | 0.3257±0.0903 |
| Scratch$_{NAS}$5 | 0.0367±0.0314 | 0.0196±0.0123 | 0.3258±0.1006 |
| Scratch$_{NAS}$6 | 0.0333±0.0205 | 0.0294±0.0177 | 0.2953±0.0808 |
| VGG11$_{NAS}$1 | 0.0249±0.0203 | **0.0399±0.0289** | **0.1559±0.0950** |
| VGG11$_{NAS}$2 | 0.0342±0.0242 | 0.0495±0.0303 | 0.1793±0.1101 |
| VGG11$_{NAS}$3 | 0.0504±0.0397 | 0.0554±0.0300 | 0.1706±0.1034 |
| VGG11$_{NAS}$4 | 0.0677±0.0531 | 0.0854±0.0544 | 0.1814±0.1032 |
| VGG19$_{NAS}$1 | 0.0199±0.0105 | 0.0205±0.0199 | 0.1868±0.0933 |
| VGG19$_{NAS}$2 | 0.0082±0.0053 | **0.0244±0.0222** | **0.1728±0.0826** |
| VGG19$_{NAS}$3 | 0.0165±0.0136 | 0.0322±0.0276 | 0.1937±0.0748 |
| VGG19$_{NAS}$4 | 0.0159±0.0094 | 0.0342±0.0225 | 0.2067±0.0854 |
| Scratch$_{DEL}$2 | 0.0235±0.0181 | 0.0131±0.0068 | 0.3123±0.0922 |
| Scratch$_{DEL}$3 | 0.0193±0.0184 | **0.0149±0.0072** | **0.3118±0.0886** |
| Scratch$_{DEL}$4 | 0.0199±0.0173 | 0.0145±0.0075 | 0.3157±0.0891 |
| Scratch$_{DEL}$5 | 0.0219±0.0192 | 0.0154±0.0082 | 0.3177±0.0887 |
| Scratch$_{DEL}$6 | 0.0224±0.0184 | 0.0178±0.0122 | 0.3116±0.0864 |
| VGG11$_{DEL}$2 | 0.0235±0.0221 | **0.0436±0.0274** | **0.1674±0.1039** |
| VGG11$_{DEL}$3 | 0.0329±0.0270 | 0.0482±0.0274 | 0.1675±0.1041 |
| VGG11$_{DEL}$4 | 0.0447±0.0369 | 0.0595±0.0383 | 0.1703±0.1020 |
| VGG19$_{DEL}$2 | 0.0112±0.0071 | **0.0216±0.0198** | **0.1791±0.0871** |
| VGG19$_{DEL}$3 | 0.0052±0.0045 | 0.0176±0.0200 | 0.1857±0.0662 |
| VGG19$_{DEL}$4 | 0.0062±0.0038 | 0.0201±0.0191 | 0.1920±0.0678 |
| Scratch$_{Best}$ | 0.0212±0.0195 | **0.0129±0.0078** | **0.3007±0.0941** |
| VGG11$_{Best}$ | 0.0374±0.0297 | **0.0496±0.0496** | **0.1603±0.0985** |
| VGG19$_{Best}$ | 0.0112±0.0071 | **0.0216±0.0198** | **0.1791±0.0871** |

Table 6: Results on the second cross-validation partition of the CT misalignments dataset

| Model | Training Absolute Error (mm) | Validation Absolute Error (mm) | Test Absolute Error (mm) |
|---|---|---|---|
| VGG11 | 0.0791±0.0475 | 0.1204±0.0806 | 0.1108±0.0437 |
| VGG19 | 0.0325±0.0274 | **0.0638±0.0574** | **0.0796±0.0535** |
| Scratch$_{NAS}$1 | 0.1059±0.1259 | 0.0424±0.0346 | 0.1958±0.1586 |
| Scratch$_{NAS}$2 | 0.1132±0.1189 | **0.0520±0.0501** | **0.1914±0.1625** |
| Scratch$_{NAS}$3 | 0.1164±0.1262 | 0.0548±0.0649 | 0.2335±0.1858 |
| Scratch$_{NAS}$4 | 0.1194±0.0907 | 0.0883±0.0486 | 0.2085±0.1329 |
| Scratch$_{NAS}$5 | 0.1346±0.1304 | 0.0916±0.0602 | 0.2270±0.1601 |
| Scratch$_{NAS}$6 | 0.1485±0.1252 | 0.1070±0.0648 | 0.2488±0.1779 |
| VGG11$_{NAS}$1 | 0.0099±0.0068 | 0.0141±0.0142 | 0.0344±0.0230 |
| VGG11$_{NAS}$2 | 0.0146±0.0129 | 0.0152±0.0112 | 0.0335±0.0163 |
| VGG11$_{NAS}$3 | 0.0126±0.0114 | **0.0158±0.0154** | **0.0293±0.0108** |
| VGG11$_{NAS}$4 | 0.0101±0.0100 | 0.0173±0.0183 | 0.0312±0.0187 |
| VGG19$_{NAS}$1 | 0.0344±0.0204 | 0.0660±0.0351 | 0.0441±0.0257 |
| VGG19$_{NAS}$2 | 0.0220±0.0180 | 0.0676±0.0541 | 0.1279±0.1256 |
| VGG19$_{NAS}$3 | 0.0523±0.0299 | **0.0719±0.0404** | **0.0378±0.0369** |
| VGG19$_{NAS}$4 | 0.0704±0.0366 | 0.0961±0.0364 | 0.0494±0.0280 |
| Scratch$_{DEL}$2 | 0.1091±0.1193 | **0.0466±0.0420** | **0.1927±0.1605** |
| Scratch$_{DEL}$3 | 0.1081±0.1207 | 0.0474±0.0485 | 0.2066±0.1702 |
| Scratch$_{DEL}$4 | 0.1074±0.1097 | 0.0610±0.0471 | 0.2063±0.1586 |
| Scratch$_{DEL}$5 | 0.1150±0.1141 | 0.0694±0.0499 | 0.2119±0.1592 |
| Scratch$_{DEL}$6 | 0.1221±0.1151 | 0.0785±0.0531 | 0.2213±0.1636 |
| VGG11$_{DEL}$2 | 0.0118±0.0086 | 0.0136±0.0112 | 0.0311±0.0195 |
| VGG11$_{DEL}$3 | 0.0065±0.0053 | 0.0126±0.0126 | 0.0297±0.0134 |
| VGG11$_{DEL}$4 | 0.0066±0.0053 | **0.0136±0.0141** | **0.0287±0.0174** |
| VGG19$_{DEL}$2 | 0.0260±0.0109 | 0.0670±0.0390 | 0.0804±0.0689 |
| VGG19$_{DEL}$3 | 0.0343±0.0207 | 0.0701±0.0274 | 0.0596±0.0478 |
| VGG19$_{DEL}$4 | 0.0439±0.0260 | **0.0760±0.0321** | **0.0556±0.0415** |
| Scratch$_{Best}$ | 0.1121±0.1145 | **0.0467±0.0421** | **0.2005±0.1416** |
| VGG11$_{Best}$ | 0.0081±0.0077 | **0.0148±0.0163** | **0.0283±0.0155** |
| VGG19$_{Best}$ | 0.0418±0.0250 | **0.0706±0.0284** | **0.0281±0.0312** |

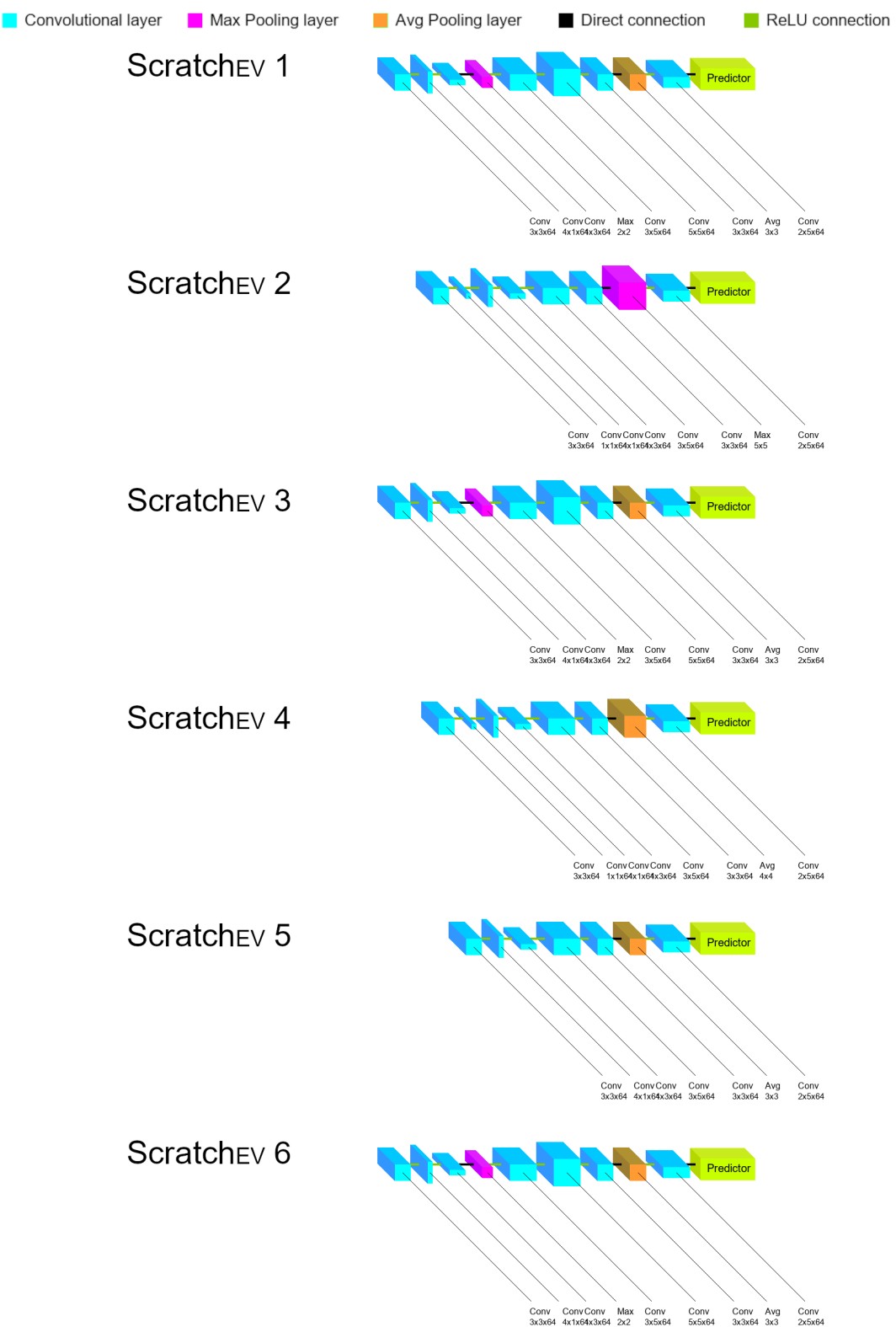

Figure 3: Architectures of the six Scratch$_{NAS}$ generated in the first cross-validation partition.

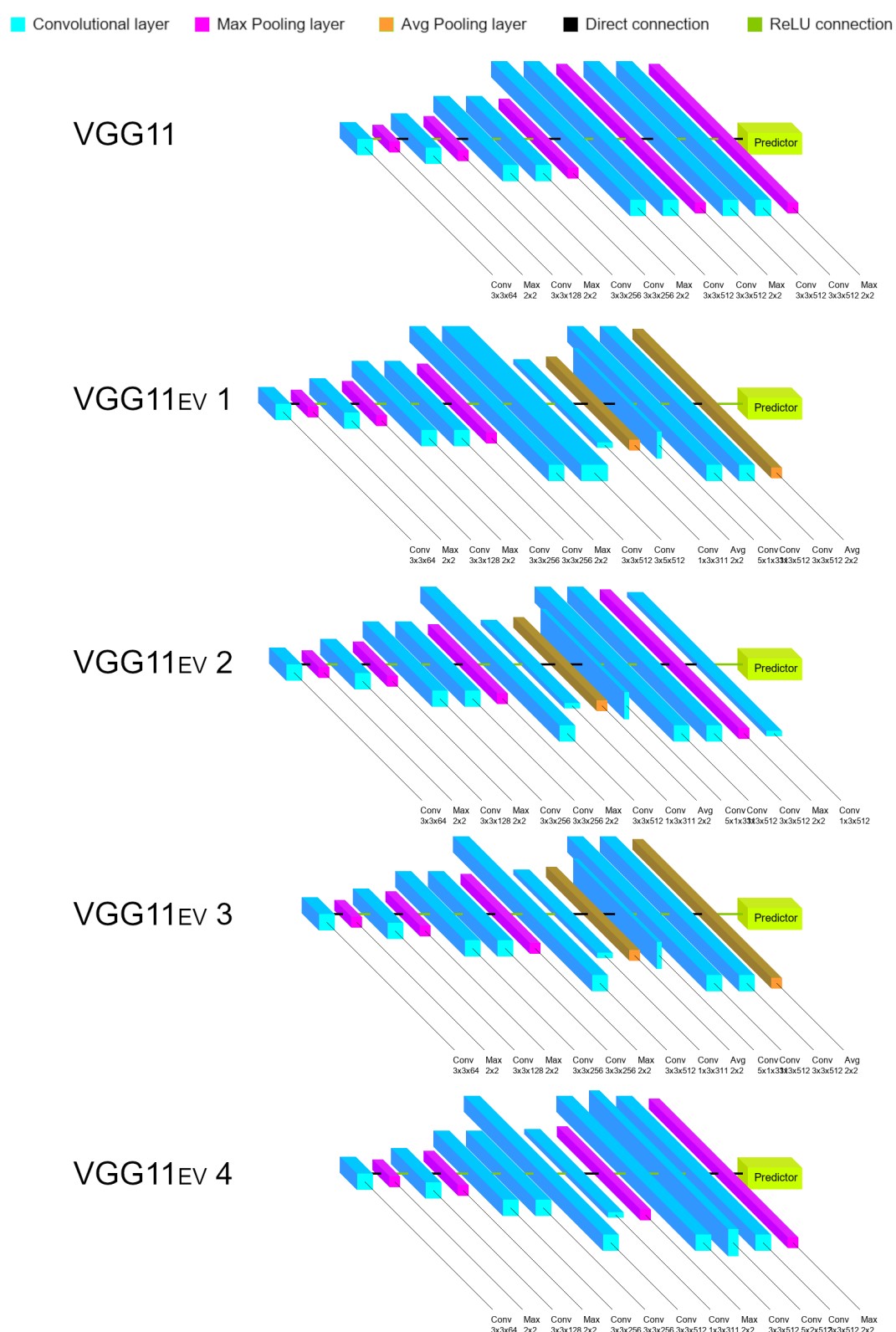

Figure 4: Architectures of VGG11 and the four VGG11$_{NAS}$ generated in the first cross-validation partition.

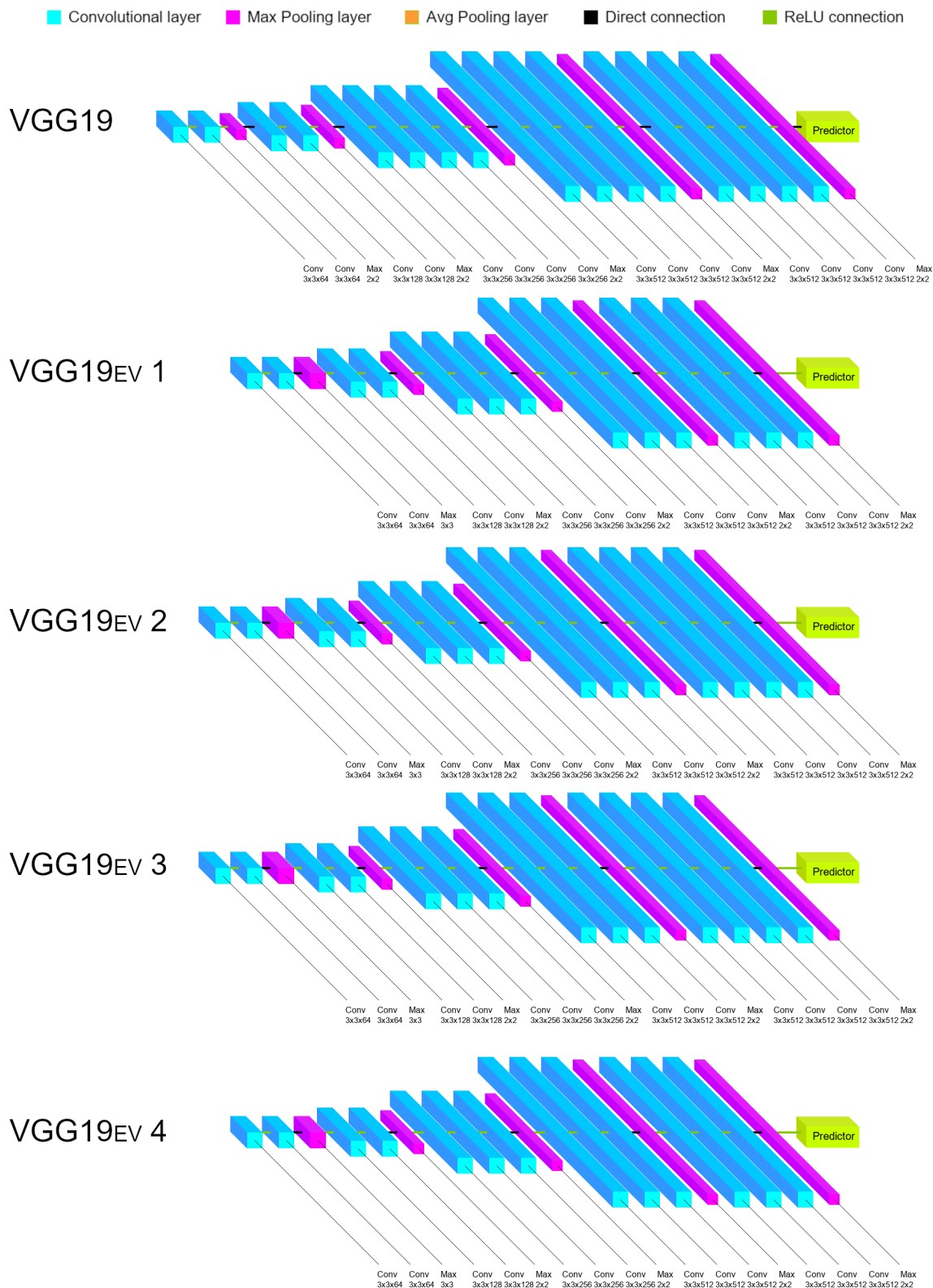

Figure 5: Architectures of VGG19 and the four VGG19$_{\text{NAS}}$ generated in the first cross-validation partition.

