# OpenReview forum: "Improving Transfer Learning by means of Ensemble Learning and Swarm Intelligence-based Neuroevolution"
_automl.cc/AutoML/2024/Conference — AutoML 2024_

### Official Review · Reviewer_KnbD · 2024-03-25

**Potential Impact On The Field Of Automl Rating:** 2
**Technical Quality And Correctness Rating:** 1
**Clarity Rating:** 2

**Summary Of Contributions:**

This work proposes evolutionary NAS and NAS+DEL algorithms and analyzes them in the context of a simple standard baseline (CIFAR10) and a biomedical transfer learning task.

**Actions Required To Increase Overall Recommendation:**

The largest flaw that impacted my score is the final layer issue mentioned previously. However, fixing this (assuming I understood the experimentation correctly) will require most (if not all) of the experiments to be rerun, which may not be possible during the rebuttal period. Fixing the other issues mentioned previously will only increase my score by up to 1 point. Otherwise, if I am misinterpreting the paper and the final layer issue is not actually a problem, then please carefully clarify this in the main paper.

EDIT: I appreciate the large changes made in response to my and other reviews. I have edited my score from "reject" to "borderline leaning accept".

**Clarity:**

The work is only somewhat clear overall.

Is there a difference between "models" (lines 144 and 152) and "employed bees" (lines 146 and 154)?

Scenario 2 is not fully explained. What exactly is the problem, and how is a model helping to solve it? You seem to be predicting the value of misalignment from each simulated image, but this is not clear. What is the practical use of this?

What is the difference between "shallow searches" and "deep search" in Table 2? Is this referring to depth of the architecture, or hyperparameters of the search algorithm? This is not explained in the main text nor in the table caption.

The tables of results (Tables 3-6) are generally quite dense and hard to draw conclusions from. Could this data be visualized?

**Overall Review:**

The introduction is currently very long. If it is split into a shorter introduction and a related works and/or background section, your readers can navigate it more easily.

As all algorithms are in the appendix, please describe each one (at a high level) more in your methods section. What the algorithms are doing is much more important to include in the main paper than, for example, the hyperparameters and data preprocessing (which are details usually put in the appendix).

Minor comments (not counted negatively):
* The citation style seems odd. Use `\citep` for parenthetical citations, rather than manually adding square brackets.
* The writing switches between past tense ("was", "were") and present tense ("is", "are"). Try to be consistent with the tense.
* When linking to a URL, please put the full URL rather than hiding it with text.
* Line 59: "...differ the greatest..." -> "...differ greatly..."
* Line 71: "...with the use DEL..." -> "...with the use of DEL..."
* Line 110: "The performance of DEL, NAS, and its combination was..." -> "The performances of DEL, NAS, and their combination were..."
* Line 151: "...tests we performed..." -> "...tests were performed..."
* Figure 1: there is no red indication as mentioned
* Table 2: "Comuting time" -> "Computing time"

**Potential Impact On The Field Of Automl:**

This is one of the first works (to my limited knowledge) to analyze the combination of NAS, DEL, and transfer learning outside of standard computer vision baselines. However, there are some experimental flaws mentioned below that limit the experimental validity.

**Review Confidence:**

3

**Review Rating:**

6

**Review Summary:**

This work proposes some algorithms that may be useful and impactful, but the experimental flaws hinder the validity of the claims made.

**Technical Quality And Correctness:**

When applying VGG11 (or VGG19), it is standard to change the dimension of the output layer from 1000 to the either the number of classes (10) or a single regression value (and reinitialize this layer) rather than add another layer. The latter approach, so done in this work, requires the model to classify the images or compute the regression value beginning with a very limited feature set of Imagenet classification scores, rather than the much higher dimension and likely more useful feature set from the layer before. This is a major experimental flaw that specifically hinders the results of the baseline models more than the searched models, as the latter benefits from architecture changes with parameter reinitialization.

Further, the reported CIFAR10 accuracies seem rather low, especially for VGG11. For how many epochs were these models trained? Was validation error converging? A potential confounding variable is that the Chimera (and Gaggle) algorithms seem to continually train models as they undergo mutations, which leads to overall longer training times and thus higher accuracies. This seems related to the primacy bias (Nikishin et al., 2022).
* Nikishin, E., Schwarzer, M., D’Oro, P., Bacon, P. L., & Courville, A. (2022, June). The primacy bias in deep reinforcement learning. In International conference on machine learning (pp. 16828-16847). PMLR.

I am a bit confused by your use of cross-validation. This is typically used to select the best design settings (or in this case a single best architecture), then that architecture can be retrained from scratch on all training and validation data, leaving only the test data out. Doing so would streamline your work, as it is currently quite messy with many models and accuracies reported in each table.

---

### Official Review · Reviewer_YCQ3 · 2024-03-27

**Potential Impact On The Field Of Automl Rating:** 2
**Technical Quality And Correctness Rating:** 3
**Clarity Rating:** 3
**Actions Required To Increase Overall Recommendation:** Provide reproducible code. Improve th…

**Summary Of Contributions:**

The paper examines the combination of neural architecture search and ensembling. It does so on two applications - a standard data set, as well as a rodent imaging data set. It uses an in-house NAS method, combining it with ensembling for more robust results.

**Clarity:**

The paper is mostly clearly written.
However, the methodology would have been easier to follow if the descriptions of the different approaches used had been broken out into e.g. bullet points, maybe with bolded key words, to make it easier to see where each model was described.

In table 2, why is the computing time for the base model longer than the ensemble?

Additionally,  couple of smaller changes would make it easier to follow:
- Write out the addresses of links, so the paper can be read in a printed version as well.
- Use bold text or some other method to guide the reader to the most interesting parts of the results tables.
- Tables 1 and 3 have the caption above the table, but table 2 has is underneath.
- No sum of computational resources used is provided. This would make it easier for the reader (e.g. all of the results presented here required X amount of compute time).
- Line 142: consisted *of*?

**Overall Review:**

The authors wrote a thoughtful ethics statement, which is a big plus. They also reflected about the limitations of the paper (second to last paragraph).

**Potential Impact On The Field Of Automl:**

The motivation is to provide more accurate and robust models for small medical imaging data sets. It provides a small step in that direction, but if in the right direction is still useful.

**Reproducibility:**

This is a major weakness of the paper. The requirements file is filled with references to local packages, also including a user name. The .yml file also includes a user name. Additionally, I could not find the data or code for the second set of experiments.

**Review Confidence:**

3

**Review Rating:**

6

**Review Summary:**

The paper includes a thoughful consideration of its limitations and ethical implications, and brings in a less-studied and important application. However, I have concerns about the reproducibility, as well as finding some of the paper hard to follow (see clarity).

Edit: I have updated my rating from 'Weak Reject' to 'Borderline Leaning Accept' following the author responses.

**Technical Quality And Correctness:**

The experiments are run on specific scenarios and so cannot answer questions about NAS and ensembles in general. But these limitations are clearly described by the authors.

---

### Official Review · Reviewer_wWgY · 2024-03-28

**Potential Impact On The Field Of Automl Rating:** 1
**Technical Quality And Correctness Rating:** 2
**Clarity Rating:** 1

**Summary Of Contributions:**

The paper  proposes a combination of Deep Ensemble Learning (DEL) and Neural Architecture Search (NAS) for small datasets, where NAS typically overfits. On two types of tasks the combination of DEL and an in-house NAS optimizer improves results compared to  stand-alone models.

**Actions Required To Increase Overall Recommendation:**

I would need to be convinced that there is a substantial technical contribution.

**Clarity:**

The paper reads more like a case study than a paper with a substantial methodological contribution. Specifically, Section 2 describes the experimental setup, but the NAS method (Chimaera) and the ensembling techniques are not. The proposed algorithm and its technical components need a thorough description, and I do not currently believe there is one.

 Moreover, the tables are dense with metrics, most of which seem unnecessary. Please highlight important takeaways in the plots, and benchmark against competitive NAS methods.

**Overall Review:**

The paper is primarily a case study on how one specific NAS method produces questionable results through presumed overfitting, and how DEL in conjunction with the specific NAS method yields improvement over the two in isolation. Thus, the learnings are thus very niche. Unfortunately, I do not see there being a substantial technical contribution (e.g. the development of a novel algorithm or fundamental learnings related to NAS or DEL).

The paper lacks clarity in multiple areas, such as the technical description of the method, and the empirical results.

**Potential Impact On The Field Of Automl:**

To me, the impact of this paper is low. Ensembling of models is known to be an effective tool when model predictions are diverse. Thus, the paper shows that the fact holds true in the combination of overfit NAS as well.

**Review Confidence:**

3

**Review Rating:**

3

**Review Summary:**

See above.

**Technical Quality And Correctness:**

The in-house NAS method, which is not thoroughly described, is thus presumed to overfit throughout the paper. Given that there is not sufficient information on the NAS method itself, I wonder why there is not more focus on why overfitting of the NAS method on it occurs in the first place. Moreover, the ensembling techniques are not formally described. Thus, it is very difficult do assess whether there is a technical contribution _at all_, and if so, what it entails from an algorithm perspective.


I will say, there is a possibility that I have misunderstood an important dynamic of the method.

---

### Official Review · Reviewer_A1iX · 2024-03-31

**Potential Impact On The Field Of Automl Rating:** 3
**Technical Quality And Correctness:** 1.  The combination of DEL and NAS, p…
**Technical Quality And Correctness Rating:** 3
**Clarity:** 1. The paper predominantly focuses on…
**Clarity Rating:** 3
**Actions Required To Increase Overall Recommendation:** Please address the weaknesses identif…

**Summary Of Contributions:**

The paper presents a comprehensive study on enhancing transfer learning models using Deep Ensemble Learning (DEL) and Neural Architecture Search (NAS) driven by swarm intelligence. It addresses the challenges faced by NAS in very small or complex datasets, such as overfitting during validation, by integrating DEL to mitigate bias and variance in predictions. The study evaluates the combined approach of NAS and DEL on two distinct scenarios: CIFAR-10 image classification and misalignment regression in CT slices with artifacts. It demonstrates that ensembles, particularly when combined with NAS, significantly outperform individual models and pre-designed architectures, especially in handling small or complex datasets typical in medical imaging applications. This is achieved by promoting model diversity and focusing on specific parts of the task through ensemble methods, effectively reducing the computational expenses while maintaining prediction complexity.

**Overall Review:**

The paper "Improving Transfer Learning by means of Ensemble Learning and Swarm Intelligence-based Neuroevolution" presents an intriguing approach to enhance transfer learning by integrating Deep Ensemble Learning (DEL) and Neural Architecture Search (NAS), particularly focusing on the context of small and complex datasets. The paper is heavily empirical, it builds on established theories of ensemble learning, transfer learning, and neural architecture search. That said many sections require additional work.

If the weakness identified are resolved, this can become an accept!

**Potential Impact On The Field Of Automl:**

The method's focus on enhancing transfer learning for small and complex datasets directly tackles a prevalent issue in AutoML. By demonstrating improved performance in such challenging scenarios, the paper provides valuable insights and a potential framework for future research aimed at broadening the applicability of AutoML techniques.

By exploring the synergies between ensemble learning and architecture search, the paper contributes to a deeper understanding of how these methods can be combined to enhance model performance. This contribution is likely to inspire further research into novel ensemble and architecture search strategies within the AutoML domain.

**Review Confidence:**

4

**Review Rating:**

8

**Review Summary:**

More clarity for the experiments conducted are required to provide a comprehensive review.

---

### Meta-Review · Area_Chair_zB4Z · 2024-04-20

**Paper Recommendation:** Accept
**Confidence:** 4

**Metareview:**

The authors did a good job responding to the reviews. While there's still some disagreement among the reviewers, the positives now outweight the negatives.

---

### Decision · Program_Chairs · 2024-04-29

**Decision:**

Accept

**Comment:**

Thank you for submitting your paper. We are happy to tell you that we accept your paper to the main track. See you in Paris.